# Feasibility and benefits of joint learning from MRI databases with different brain diseases and modalities for segmentation

**Wentian Xu**[*][1]                                                                 WENTIAN.XU@ENG.OX.AC.UK
**Matthew Moffat**[*][1]                                                             MATTHEW.MOFFAT@ENG.OX.AC.UK
**Thalia Seale**[1]                                                                  THALIA.SEALE@ENG.OX.AC.UK
**Ziyun Liang**[1]                                                                   ZIYUN.LIANG@ENG.OX.AC.UK
**Felix Wagner**[1]                                                                  FELIX.WAGNER@ENG.OX.AC.UK
**Daniel Whitehouse**[2]                                                             DW555@CAM.AC.UK
**David Menon**[2]                                                                   DKM13@CAM.AC.UK
**Virginia Newcombe**[2]                                                            VFJN2@CAM.AC.UK
**Natalie Voets**[3]                                                                 NATALIE.VOETS@NDCN.OX.AC.UK
**Abhirup Banerjee**[1]                                                             ABHIRUP.BANERJEE@ENG.OX.AC.UK
**Konstantinos Kamnitsas**[1]                                                        KONSTANTINOS.KAMNITSAS@ENG.OX.AC.UK

[1] *Department of Engineering Science, University of Oxford, Oxford, United Kingdom*

[2] *Department of Medicine, University of Cambridge, Cambridge, United Kingdom*

[3] *Nuffield Department of Clinical Neurosciences, University of Oxford, Oxford, United Kingdom*

**Editors:** Accepted for publication at MIDL 2024

## Abstract

Models for segmentation of brain lesions in multi-modal MRI are commonly trained for a specific pathology using a single database with a predefined set of MRI modalities, determined by a protocol for the specific disease. This work explores the following open questions: Is it feasible to train a model using multiple databases that contain varying sets of MRI modalities and annotations for different brain pathologies? Will this joint learning benefit performance on the sets of modalities and pathologies available during training? Will it enable analysis of new databases with different sets of modalities and pathologies? We develop and compare different methods and show that promising results can be achieved with appropriate, simple and practical alterations to the model and training framework. We experiment with 7 databases containing 5 types of brain pathologies and different sets of MRI modalities. Results demonstrate, for the first time, that joint training on multi-modal MRI databases with different brain pathologies and sets of modalities is feasible and offers practical benefits. It enables a single model to segment pathologies encountered during training in diverse sets of modalities, while facilitating segmentation of new types of pathologies such as via follow-up fine-tuning. The insights this study provides into the potential and limitations of this paradigm should prove useful for guiding future advances in the direction. Code and pretrained models: https://github.com/WenTXuL/MultiUnet

## 1. Introduction

Segmentation of pathologies in Magnetic Resonance Imaging (MRI) of the brain plays a crucial role in disease diagnosis and treatment. Deep learning has marked a significant leap in medical image segmentation. Existing works for segmentation of brain lesions in MRI

---

[*] Contributed equally

(Kamnitsas et al., 2017; Isensee et al., 2021; Chen et al., 2019b), however, are commonly trained to segment a specific pathology, using a database acquired with a predefined set of imaging modalities determined by a protocol for the specific disease. One database may contain several MRI modalities, but the standard methods require the same set of modalities available during training and inference. Therefore databases with diverse sets of modalities are not jointly leveraged. Performance of a model trained on a single database is heavily influenced by the database's size. Instead, learning from multiple databases could enhance generalization, while training using varying modality sets and pathologies could enable segmenting different sets of modalities and pathologies with a single model. Thus, exploring how to train models using multiple databases with heterogeneous modalities is a promising avenue. Towards this goal, this study investigates the following open questions:

**Q1:** Can we develop a lesion segmentation model that can learn from different multi-modal MRI databases when each has a different set of modalities and pathologies?

**Q2:** Can databases with different sets of modalities mutually benefit via joint training?

**Q3:** Can training on different databases help the model segment pathologies that have been seen during training but on sets of modalities unlike those used in training?

**Q4:** How will the jointly trained model perform when encountering pathologies not presented during its initial training, either directly or through subsequent fine-tuning?

Several studies have previously investigated the concept of joint training on multiple medical databases. (Yan et al., 2020) and (Ulrich et al., 2023) trained a model on multiple databases for lesion detection and structure segmentation, but they only trained on one modality (CT). (Moeskops et al., 2016) trained a model on databases of different anatomies in T1 and CT but did not explore multi-modal brain MRI. Recently, (Wang et al., 2023) built a foundation model for 3D medical images based on SAM (Kirillov et al., 2023), trained on a large number of medical databases with different modalities using a large transformer as backbone. That model can only process a single modality at a time, it does not learn relations between modalities, and it requires the user to click or draw a box around the object for segmentation. Similar is the training of UniverSeg (Butoi et al., 2023), a model that handles one modality at a time, and requires manual segmentations given at inference-time to segment a new database. In contrast, our work is specifically focused on finding effective methods for handling diverse sets of multi-modal MRI data, without manual inputs.

Our study also relates to works that focused on the setting where a single training database is available with specific modalities, aiming to reduce performance degradation when modalities are missing at test time (Havaei et al., 2016; Hu et al., 2020; Islam et al., 2021; Zhou et al., 2021). Some such works (Islam et al., 2021; Zhou et al., 2021) require introduction of complex auxiliary components, such as generative models. Others enable the segmentation model to embed inputs even if modalities are missing, such as HeMIS (Havaei et al., 2016). Our study draws inspiration from these but focuses on how to handle multiple, diverse training databases with different sets of MRI modalities and pathologies.

Also related are works that studied how to segment new pathologies not seen during training. Examples are domain generalization works leveraging information from other pathologies (Gu et al., 2021), or transferring pre-trained knowledge to segment new database (Chen et al., 2019a), or unsupervised methods (Pinaya et al., 2022a,b) that can segment pathologies without training explicitly for it. Our work instead explores the feasibility of approaching this problem with a model jointly-trained on heterogeneous sets of modalities.

The main contributions of this study can be summarized as follows:

- We explore and demonstrates the feasibility and benefits of jointly training with databases with heterogeneous sets of MRI modalities and types of brain lesions.
- We explore several methods for this challenge and show promising results with appropriate, simple (thus practical) alterations to the model and training strategy.
- We approach the stated open questions (Q1-4) by conducting experiments with 7 databases in multiple settings, including segmentation on diverse sets of modalities, and segmenting types of pathologies seen and unseen during training, thus providing valueable insights into the properties of this unexplored training paradigm.

## 2. Materials and methods

### 2.1. Handling varying modalities with Unet and modality drops: Multi-Unet

Can we enable the well-established Unet to learn from databases with distinct sets of MRI modalities (Tab. 1) without extensive alterations, to retain its practicality? For this, we use a Residual Unet (Zhang et al., 2018) as base model (Fig. 1) and modify it as follows.

**Handling diverse modality sets:** Assume we aim to train a model on a collection $D$ of $N$ training databases, $D = \{D_1, ..., D_N\}$. Assume database $D_i$ has set of MRI modalities $\mathbb{M}_i$, where $C_i = |\mathbb{M}_i|$ their number. To process inputs *solely* from database $D_i$, a Unet would be designed using input layers with $C_i$ input channels. Instead, to enable a Unet to process jointly all training databases $D_i$ in $D$, with any distinct set of modalities $\mathbb{M}_i$, we design it with number of input channels $C$ equal to the number of *unique* modalities across all $N$ databases, $C = |\mathbb{M}_1 \cup ... \cup \mathbb{M}_N|$. When processing samples from databases that do not have certain modalities, the corresponding channels are filled with zeros (Fig. 1). We term the Unet trained with the above strategy as **Multi-Unet**.

**Enhancing Generalization with Random Drop of Modalities:** The above alteration enables joint training on MRI databases with diverse modality sets. This model, however, may only learn to segment specific combinations of modalities $\{\mathbb{M}_1, ..., \mathbb{M}_N\}$, those available in the training databases. It may also associate the specific combination of modalities $\mathbb{M}_i$ in database $i$ with the specific pathology annotated in database $i$, unable to segment it if given another combination. Our desiderata, instead, is to train a model that can segment any arbitrary modality combination at test time. To enable this, we drop modalities when training. For each training sample from database $D_i$ with $C_i$ number of modalities, we sample an integer number $n$ of modalities to drop from the range $[0, C_i - 1]$. Then we randomly choose $n$ out of the $C_i$ modalities of that sample and replace them with blank images. This enforces the model to segment each pathology under diverse sets of modalities and encourages feature extraction from all modalities rather than the most prevalent.

### 2.2. Embedding each modality separately: LFUnet and MAFUnet

We develop and study two alternative, more complex methods that employ separate embeddings per modality. These are inspired by previous works that focused on training with a single database while aiming to enhance generalization when modalities are missing at test time, as opposed to our study that targets learning from databases with different sets of modalities. Inspired by HeMIS (Havaei et al., 2016) that uses a pathway per

| Databases | PD | FLAIR | SWI | T1 | T1c | T2 | DWI |
|---|---|---|---|---|---|---|---|
| BRATS | | ✓ | | ✓ | ✓ | ✓ | |
| MSSEG | ✓ | ✓ | | ✓ | ✓ | ✓ | |
| ATLAS | | | | ✓ | | | |
| TBI | | ✓ | ✓ | ✓ | | ✓ | |
| WMH | | ✓ | | ✓ | | | |
| ISLES | | ✓ | | ✓ | | ✓ | ✓ |
| Tumor_2 | | | | ✓ | | | |

Table 1: Modalities in each database

Figure 1: Unet training with modality dropping.

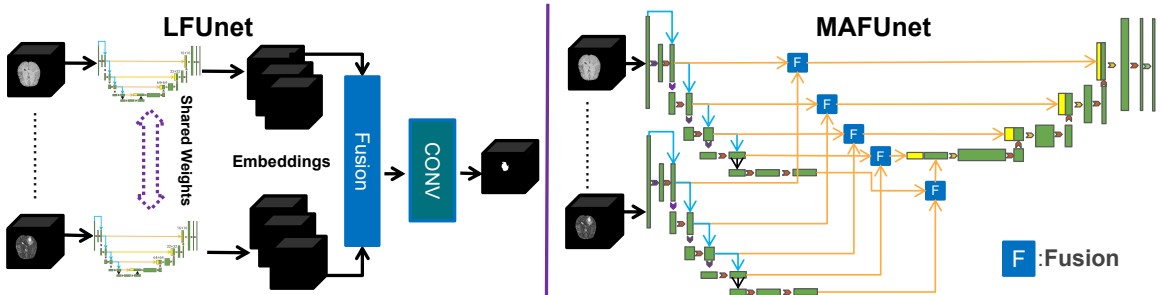

Figure 2: Architectures that embed each modality separately: LFUnet and MAFUnet

modality and fuses them by outputting their *average* and *variance*, we create a Unet-based variant that we term Late-Fusion-Unet (**LFUnet**), Fig. 2, that applies a Unet per modality and fuses their embeddings at the penultimate layer. Same Unet weights are shared to avoid overfitting. In early experiments, we found that results are improved by using fusion via weighted averaging with attention weights, rather than the averaging and variance used in HeMIS. For $C$ total possible modalities, and $\mathbf{z}_i \in \mathbb{R}^{H \times W \times D}$ the embedding of i-th modality with H,W,D spatial dimensions, the attention weights are $\mathbf{a} = softmax(conv(conv([\mathbf{z}_1, ..., \mathbf{z}_C]))) \in \mathbb{R}^{H \times W \times D \times C}$, where softmax normalizes attention across C modalities. Fusion's output is then the dot product of attention with the concatenated embeddings, $\mathbf{z}_{fused} = \mathbf{a} \cdot [\mathbf{z}_1, ..., \mathbf{z}_C]$. LFUnet requires high memory and computation due to one Unet replica per modality. We therefore developed a further variant, termed Multi-scale Attention Fusion Unet (**MAFUnet**). This uses an encoder per modality and fuses embeddings at each scale, as shown in Fig. 2. Modality dropping (Sec. 2.1) is also employed in these two methods.

## 2.3. Databases

Seven brain MRI databases are used, including 484 tumor cases from **BRATS** 2016 (Bakas et al., 2017), 53 Multiple Sclerosis cases from **MSSEG** 2016 (Commowick et al., 2018), 655 stroke lesion cases from **ATLAS** v2.0 (Liew et al., 2022), 281 traumatic brain injury cases (**TBI**) from our institutions, 60 White Matter Hyperintensity cases from the **WMH** challenge (Kuijf et al., 2022), 28 stroke lesion cases from SISS subtask of **ISLES** 2015

(Maier et al., 2017), and another tumor database of 51 cases collected at our institutions (**Tumor_2**). Table 1 shows modalities available in each database. For training we use 444, 459, 37, 156, and 42 cases from BRATS, MSSEG, ATLAS, TBI, and WMH respectively, with remaining cases used for evaluation. ISLES and Tumor_2 are used for evaluation only. **Preprocessing:** We skull-strip, resample to 1×1×1 mm and apply z-score intensity normalization to all data. All lesion classes are merged in a single 'lesion' label for all experiments.

## 3. Experiments and Results

Learning from brain lesion MRI databases with heterogeneous modality sets has not been explored before. Thus we firstly set to identify promising models among those developed (Sec. 2) and then use them to assess the benefits of learning from diverse databases. **Training setup:** To address class imbalance during training we over-sample small databases to the frequency of the largest. We train with batch size of 2 for 600 epochs with learning rate 0.001, decayed to 0.0001 after 150 epochs.

### 3.1. Which architecture is more robust to missing modalities?

Our ultimate desiderata is to develop a model that can learn from databases with heterogeneous modality sets and can segment new data with modality sets that differ from those seen during training. A first step towards this is to identify architectures that perform best on a single database when modalities are dropped. We show such experiments in Table 2. We find that to reduce performance degradation with missing modalities, training with modality dropping is essential. In this training setting, the most favourable models are MAFUnet and Multi-Unet, all three achieve similar performance when all modalities are present, but LFUnet degrades the most with missing modalities, while requiring highest memory. Therefore in followup experiments we focus on MAFUnet and Multi-Unet.

Table 2: **Performance when training on single database and robustness to missing modalities:** We train each model on each of the five databases separately (BRATS, TBI, MSSEG, ATLAS, WMH) in two ways: using all available modalities during training (top) or randomly dropping modalities (bottom). We then evaluate each model on the corresponding database using all available modalities, and report average Dice over all databases (out of brackets). To evaluate robustness to missing modalities, we test each model with all possible combinations of modalities (dropping 1 or more) on each database , and report how much Dice decreased on average across all databases and all combinations of dropped modalities, in comparison to using all modalities (in brackets). Training with modality drop is necessary to improve performance with missing modalities, therefore we concentrate on bottom row. Therein, all architectures achieve similar Dice with all modalities availble but LFUnet drops the most with missing modalities. Hence Multi-Unet or MAFUnet are preferred.

| Average Dice and Difference | LFUnet | MAFUnet | Multi-Unet |
|---|---|---|---|
| Training with all modalities | 0.665 (-0.348) | 0.653 (-0.333) | 0.658 (-0.528) |
| Training with modality dropping | 0.650 (-0.194) | 0.640 (**-0.123**) | 0.649 (**-0.125**) |

### 3.2. Can we train one model on multiple databases with varying modalities?

We now set out to train a single model using 5 databases (MSSEG, TBI, WMH, BRATS, ATLAS). We assess whether it is possible to obtain one model that can segment all databases well, regardless that each has a distinct set of modalities and pathology. For this, we train Multi-Unet and MAFUnet with all 5 databases. We use each to segment all databases. We compare their performance with instances of Residual Unets trained separately on each database in Table 3. Results verify that training with multiple databases is feasible, regardless the distinct sets of modalities. The simple (thus practical) Multi-Unet is particularly effective! It gives results comparable or better than the database-specific models, with the great practical benefit that knowledge is wrapped in a single model. In following sections we focus on Multi-Unet and explore what more benefits are unlocked by such model that has condensed knowledge from varying sets of modalities and pathologies.

Table 3: **Performance when training on multiple databases:** Separate models were trained on each Single Database (SD) and evaluated on the same database, compared with models jointly trained on Multiple Databases (MD), shown either all available modalities during training (all mods) or random modality drop (drop). All available modalities are provided for testing. For each experiment, average Dice of 3 runs (different RNG seeds) is reported. Most importantly, one MD model can segment all databases regardless their different modality sets, in contrast to SD that can only segment the database seen during training. MultiUnet-MD performs best on average, showing benefits of transfer learning especially on smaller databases (MSSEG, TBI, WMH).

| Methods | MSSEG | TBI | WMH | BRATS | ATLAS | Avg |
|---|---|---|---|---|---|---|
| Unet-SD (all mods) | 0.659 | 0.515 | 0.721 | 0.908 | 0.487 | 0.658 |
| Unet-SD (drop) | 0.657 | 0.517 | 0.695 | 0.908 | 0.487 | 0.653 |
| MultiUnet-MD (drop) | 0.676 | 0.521 | 0.725 | 0.906 | 0.485 | **0.663** |
| MAFUnet-MD (drop) | 0.674 | 0.515 | 0.708 | 0.891 | 0.468 | 0.651 |

### 3.3. Segmenting "known" pathologies in databases with different modality sets

Models that can learn from diverse databases and sets of modalities could potentially unlock the practical benefit of enabling segmentation of new databases that have the same pathologies as some of the training databases but have different sets of modalities than them. Table 4 shows related experiments. The Multi-Unet previously trained on 5 databases with modality drop has only seen stroke in T1 (from ATLAS) during training. Results demonstrate that this model can use the additional modalities (T2, Flair) of ISLES to better segment stroke therein, thanks to having processed other types of pathologies in these modalities during training. The same model can also segment tumor in a new database that contains only T1 better than a model trained solely on BRATS using modality drop to enable inference on just T1. These results may not reach state-of-the-art performance on these databases, but they demonstrate the promising benefits from multi-database training when the challenge of handling distinct sets of modalities is overcome.

Table 4: **Segmenting "known" pathologies in new databases with different sets of modalities:** Stroke segmentation model trained solely on ATLAS, which contains only T1, cannot use T2 and FLAIR available in ISLES (first row). Multi-Unet trained on (ATLAS, MSSEG, WMH, TBI, BRATS) with modality drop (fourth row) has seen stroke only in ATLAS's T1 during training, but can use ISLES's T2 and FLAIR for better segmentation, as T2 and FLAIR have been seen in other training databases. The same model segments tumor in unseen database Tumor_2, which contains only T1, better than a model trained only on BRATS, which contains T1, T1c, T2, FLAIR (second row). Average Dice of 3 runs (different RNG seeds) reported.

| Inference
Training | Stroke_2 (ISLES) | | | Tumor_2 |
|---|---|---|---|---|
| | (T1) | (T1,T2) | (FLAIR, T1, T2) | (T1) |
| Stroke_1 (ATLAS) | 0.053 | N/A | N/A | |
| Tumor_1 (BRATS, drop mods) | | | | 0.646 |
| All Databases (all mods) | 0.046 | 0.064 | 0.471 | 0.538 |
| All Databases (drop mods) | **0.348** | **0.424** | **0.559** | **0.697** |

### 3.4. Can joint training of multiple databases facilitate segmentation of pathologies unseen during training, directly or after fine-tuning?

**Segment unknown pathologies:** We now explore whether learning from multiple brain lesion databases can facilitate direct segmentation of types of pathologies unseen during training. Related results shown in Table 5. We compare results with state-of-the-art unsupervised segmentation models, to compare capabilities of segmenting new type of pathology. Regardless the simplicity of Multi-Unet, which is a standard Unet trained with simple modifications, it outperforms complex recent unsupervised methods, which demonstrates the potential from this training paradigm in extracting knowledge useful for new segmentation tasks. Regardless, results are significantly below what is possible with models supervised with labels for the target pathologies. We therefore now set out to assess if these Multi-Unet instances can leverage supervision via fine-tuning for the new pathologies.

**Segmenting new pathologies after fine-tuning:** We here study if Multi-Unets pre-trained on diverse databases can benefit segmentation of new pathologies if used as starting checkpoints for fine-tuning to the target database. Table 6 shows related experiments. Results demonstrate the prior knowledge from diverse databases consistently benefits fine-tuning to new pathologies in comparison to training from scratch, both with limited and plenty of labels. We demonstrate this also with a more sophisticated fine-tuning method based on progressive nets (Rusu et al., 2016), by learning a new Unet 'column' that leverages prior knowledge via lateral connections from the pre-trained column.

## 4. Conclusion

This study investigated several models to enable, for the first time, joint training on multiple brain lesion MRI databases with heterogeneous sets of modalities. We demonstrate feasibility and benefits of such training paradigm on various tasks, including segmenting

Table 5: **Segmenting types of lesions never seen during training:** Using the 5 shown databases, 5 instances of Multi-Unet are trained, each on 4 databases, using the 5th for evaluating model's ability to segment the unseen pathology. Dice reported for the held-out database. Multi-Unet outperforms unsupervised models (Transformer VQ-VAE (Pinaya et al., 2022b) and DDPM (Pinaya et al., 2022a)) that have not seen these pathologies, demonstrating the benefits of learning from diverse databases. VQ-VAE and DDPM use single modalities (FLAIR, except for Atlas that only includes T1). Hence we report results of Multi-Unet when tested with the same inputs, and with all possible modalities. Results by Unet-SD trained on each target database (Tab. 3) are shown as upper bound for reference.

| Methods | Test mod. | **BRATS** | **WMH** | **TBI** | **ATLAS** | **MSSEG** |
|---|---|---|---|---|---|---|
| Unet-SD (upper bound) | all | 0.908 | 0.721 | 0.515 | 0.487 | 0.659 |
| Transf. VQ-VAE | Flair | 0.537* | 0.429* | - | - | - |
| DDPM | Flair or T1 | 0.469* | 0.272* | 0.253 | 0.225 | 0.121 |
| MultiUNet (drop mods) | Flair or T1 | 0.614 | 0.454 | 0.324 | **0.241** | **0.540** |
| MultiUNet (drop mods) | all | **0.718** | **0.470** | **0.334** | **0.241** | 0.326 |

Table 6: **Benefits for fine-tuning:** 5 Multi-Unets are trained on 4 out of 5 databases and evaluated on the 5th target database (0-shot). Performance does not reach supervised models trained from scratch with labels of the target database. However, knowledge acquired from the diverse prior databases can benefit fine-tuning. Standard fine-tuning of Multi-Unets out-performs models trained from scratch, both when using limited labels (50 for TBI, 20 for others) or labels of whole training split of target database. We also report results from a more advanced fine-tuning method using Progressive Nets (Progr).

| | **None** | **Limited Data** | | | **All Data** | | |
|---|---|---|---|---|---|---|---|
| Target | 0-shot | Scratch | Finetune | Progr. | Scratch | Finetune | Progr. |
| **BRATS** | 0.718 | 0.825 | 0.850 | 0.871 | 0.908 | 0.911 | 0.913 |
| **ATLAS** | 0.241 | 0.268 | 0.389 | 0.365 | 0.487 | 0.487 | 0.509 |
| **MSSEG** | 0.326 | 0.644 | 0.670 | 0.673 | 0.659 | 0.690 | 0.706 |
| **TBI** | 0.334 | 0.375 | 0.437 | 0.447 | 0.515 | 0.524 | 0.529 |
| **WMH** | 0.470 | 0.677 | 0.695 | 0.703 | 0.721 | 0.736 | 0.758 |

multiple pathologies with a single model, segmenting them in new databases and modality sets, and benefits for transfer learning to learn segmenting new pathologies. Interestingly, these benefits were realised with minimal alterations to a standard Unet, which allows practical applications. This study did not aim to raise state-of-the-art on the individual databases. Rather, it demonstrates the feasibility and benefits of this training paradigm for multi-modal brain MRI databases, which we hope will inspire further work in this direction. Future work could investigate more advanced frameworks that can extract invariant information to do segmentation on untrained pathologies and modalities without fine-tuning.

## Acknowledgments

TS and FW are supported by the EPSRC Centre for Doctoral Training in Health Data Science (EP/S02428X/1). FW is also supported by the Anglo-Austrian Society, and by an Oxford-Reuben scholarship. ZL is supported by scholarship provided by the EPSRC Doctoral Training Partnerships programme [EP/W524311/1]. NV is supported by the NIHR Oxford Health Biomedical Research Centre (NIHR203316). VN, NIHR Rosetrees Trust Advanced Fellowship, NIHR302544, is funded in partnership by the NIHR and Rosetrees Trust. The views expressed are those of the authors and not necessarily those of the NIHR, Rosetrees Trust or the Department of Health and Social Care. The authors also acknowledge the use of the University of Oxford Advanced Research Computing (ARC) facility in carrying out this work (http://dx.doi.org/10.5281/zenodo.22558).

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

## Appendix A. Additional metrics

In this section, we provide additional metrics for the experiments described in Sec. 3.3 and Sec. 3.2. We calculate the commonly used metrics of Sensitivity (i.e. Recall), Precision and Average Symmetric Surface Distance (ASSD).

Table 7 and Table 8 show these metrics for the experiments of Sec. 3.3 that were previously analysed in Table 3. Table 7 shows that on average across all databases, MultiUnet-MD (joint training on multiple databases) achieves highest Sensitivity (low FNs) and highest Precision (low FPs) in comparison to models trained on each database separately. Additionally, Table 8 shows that it also achieves the best (lowest) average ASSD. These results demonstrate the consistent improvements of joint training on multiple databases.

Table 9 and Table 10 present these metrics for the experiments described in Sec. 3.3 that investigate generalisation to new databases with new sets of modalities. These were previously analysed with the Dice metric in Table 4. The model trained with the studied learning framework ("All databases (drop mods)") achieves improved Sensitivity in all but one experiment, improved Precision in all experiments, and improved ASSD in all experiments. This analysis reinforce the findings that the studied framework is beneficial.

Table 7: **Sensitivity and Precision** of models trained on a single database (Unet-SD) with all modalities (all mods) or by dropping modalities during training (drop), in comparison to models trained jointly on all databases. Experimental settings are the same as Table 3. Sensitivity and precision are reported in the format "sensitivity/precision". On average across all databases ("Avg" column), MultiUnet-MD that is trained on all available databases with modality drop achieves highest sensitivity and precision, demonstrating the consistent benefits by this learning paradigm.

| Methods | MSSEG | TBI | WMH | BRATS | ATLAS | Avg |
|---|---|---|---|---|---|---|
| Unet-SD (all mods) | 67.9/80.9 | 66.8/67.8 | 73.7/78.9 | 92.2/92.3 | 63.1/81.5 | 72.7/80.3 |
| Unet-SD (drop) | 55.3/83.4 | 65.0/63.1 | 71.0/75.2 | 90.5/93.1 | 63.1/81.5 | 69.0/79.3 |
| MultiUnet-MD (drop) | 71.5/82.4 | 66.4/72.7 | 76.9/79.5 | 90.4/93.0 | 64.6/84.7 | **74.0/82.5** |
| MAFUnet-MD (drop) | 68.7/81.2 | 66.9/70.7 | 73.0/79.0 | 87.8/92.6 | 63.9/83.0 | 72.06/81.3 |

Table 8: **Average symmetric surface distance (ASSD)** of models trained on a single database (Unet-SD) with all modalities (all mods) or by dropping modalities during training (drop), in comparison to models trained jointly on all databases. The experiment settings are the same as Table 3. MultiUnet-MD that is trained on all available databases with modality drop achieves the best results (lowest ASSD) in all datatabases, demonstrating the consistent improvements by this learning paradigm.

| Methods | MSSEG | TBI | WMH | BRATS | ATLAS | Avg |
|---|---|---|---|---|---|---|
| Unet-SD (all mods) | 3.87 | 10.30 | 1.36 | 1.67 | 11.35 | 5.71 |
| Unet-SD (drop) | 4.49 | 11.99 | 2.08 | 1.52 | 11.35 | 6.29 |
| MultiUnet-MD (drop) | 3.71 | 10.42 | 1.44 | 1.67 | 9.25 | **5.30** |
| MAFUnet-MD (drop) | 3.72 | 11.54 | 1.56 | 1.99 | 11.07 | 5.98 |

Table 9: **Sensitivity and Precision** on segmenting "known" pathologies in new databases with different sets of modalities using one model. The experiment settings are the same as Table 4, but we calculate different metrics. Sensitivity and precision are reported in the format "sensitivity/precision".

| Training \ Inference | Stroke_2 (ISLES) | | | Tumor_2 |
|---|---|---|---|---|
| | (T1) | (T1,T2) | (FLAIR, T1, T2) | (T1) |
| Stroke_1 (ATLAS) | 9.7/44.5 | N/A | N/A | |
| Tumor_1 (BRATS, drop mods) | | | | **87.5**/62.0 |
| All Databases (all mods) | 4.5/66.8 | 5.1/27.0 | 80.0/53.2 | 64.1/69.8 |
| All Databases (drop mods) | **51.5/64.1** | **64.9/74.8** | **81.9/72.8** | 82.7/**79.7** |

Table 10: **Average symmetric surface distance(ASSD)** when segmenting "known" pathologies in new databases with different sets of modalities using one model. The experiment settings are the same as Table 4. Best (lower) in bold.

| Inference
Training | Stroke_2 (ISLES) | | | Tumor_2 |
| | (T1) | (T1,T2) | (FLAIR, T1, T2) | (T1) |
|---|---|---|---|---|
| Stroke_1 (ATLAS) | 37.16 | N/A | N/A | |
| Tumor_1 (BRATS, drop mods) | | | | 11.00 |
| All Databases (all mods) | 42.35 | 58.04 | 24.11 | 13.76 |
| All Databases (drop mods) | **25.44** | **16.36** | **14.4** | **7.28** |

## Appendix B. Fine-tuning on new databases that contain modality not available in original training data

When fine-tuning a pretrained model on a new database, the new data may include imaging modalities absent in the original training databases. For example, in experiments of Sec. 3.4 and Table 6, a model trained on ATLAS, BRATS, MSSEG and WMH but not the TBI database has not seen Susceptibility Weighted Imaging (SWI) modality during training, as it is only available in the TBI data. Therefore, when this pre-trained model is used as basis for fine-tuning on the TBI database, an input channel needs to be assigned to this new SWI modality. In this case, if there is an unused input channel, which was pre-trained for a modality that is not available in the new, target database for which the model will be fine-tuned, the new modality can be assigned to such an unused channel. During fine-tuning, the corresponding weights will be adjusted accordingly. In our example, the SWI modality is assigned to the PD channel, as the latter is not available in the TBI database. If the new database contains total number of modalities larger than the number available channels from the pre-training stage, the filters of the model's first layer can be expanded with additional, randomly initialised weights, to account for the additional channels.

## Appendix C. Details on Progressive neural networks for fine-tuning

In Section 3.4, We besides standard fine-tuning with further training the original model parameters on the new target database of interest, we also applied the framework of Progressive Networks (Rusu et al., 2016) for improved results. This original work developed this framework for continual learning, for a model to sequentially learn a new, different task. Knowledge is transferred from previous tasks to improve convergence speed. The model architecture aims to prevent catastrophic forgetting by instantiating a new neural network (a column) for each task being solved. Knowledge transfer between tasks is facilitated by lateral connections of features from the previously learned column to the new column. In this work, we use a similar framework for fine-tuning. We set two models (columns) with the same Unet architecture used in other experiments. The first column is the model pre-trained on the original databases. Its parameters are kept frozen, thereby retaining the knowledge learned from the original databases. The second column is a Unet with param-

eters initialised randomly, and will be trained using the new, target database. Following (Rusu et al., 2016), the second column leverages features extracted from the first column. To do this, activations from the layers of the first column before each downsampling (convolution kernel with stride greater than one) and upsampling layer are transferred to the second column. These features are concatenated with corresponding features in the same position in the second column, and are then processed by the following convolutional layer along with the feature activations from the previous layer of the second column. During both the fine-tuning and inference phases, images are inputted simultaneously into both columns.

