# OpenReview forum: "Feasibility and benefits of joint learning from MRI databases with different brain diseases and modalities for segmentation"
_MIDL.io/2024/Conference — MIDL 2024 Oral_

### Official Review · Reviewer_cdiz · 2024-02-27

**Confidence:** 5
**Preliminary Rating:** 4
**Final Rating:** 5

**Summary:**

This paper explores and demonstrates the feasibility and benefits of jointly training with databases with heterogeneous sets of MRI modalities and types of brain lesions. The authors demonstrate that this unexplored training paradigm is effective by conducting experiments with 7 databases in multiple settings, including segmentation on diverse sets of modalities, and segmenting types of pathologies seen and unseen during training.

**Strengths:**

(1) The authors improved the model's generalization capability by employing random dropping of modalities to handle data from different modalities
(2) The authors proposed LFUnet and MAFUnet to embed each modality separately, demonstrating the effectiveness of their joint training strategy.
(3) This paper demonstrates the feasibility and benefits of jointly training on various tasks, including segmenting multiple pathologies with a single model, segmenting them in new databases and modality sets, and benefits for transfer learning to learn segmenting new pathologies.

**Weaknesses:**

(1) The explanation of the experimental comparisons in Table 2 and Table 4 in the paper is somewhat ambiguous and difficult to understand.
(2) The framework diagrams of the two network models in Figure 2 could be improved for better clarity and completeness.
(3) The strategy of Random Dropping of Modalities is not specifically elaborated, which is a crucial point.

**Detailed Comments:**

(1) The explanation of the experimental comparisons in Table 2 and Table 4 in the paper is somewhat ambiguous and difficult to understand.
(2) The framework diagrams of the two network models in Figure 2 could be improved for better clarity and completeness.
(3) The strategy of Random Dropping of Modalities is not specifically elaborated, which is a crucial point.

**Justification Of Final Rating:**

The author has addressed all my concerns. This article is currently easy to understand. I would like to remind the authors to provide a more specific analysis of the experimental results to support the innovations proposed in the article.

**Justification Of The Preliminary Rating:**

The overall logic of this paper is clear, especially in the Experiments and Results description part. The authors propose several innovative network architectures and utilize comparative experiments and ablation experiments to demonstrate the effectiveness of the joint training strategy.

**Questions To Address In The Rebuttal:**

(1) Details still need more attention, such as the font size of the chart should be adjusted to an appropriate size.
(2) In the METHODOLOGY part of the paper, I suggest that the authors introduce the overall framework first, and then explain each module.
(3) I think that adding some colors to Figure 2 will make it more beautiful and clear.

---

> ### Author Response · Authors · 2024-03-17
> **Reponse to comments of R3 (#cdiz)**
>
> We are thankful to the reviewer for their work on this review and we are glad they appreciate the value of this study. We are glad the reviewer acknowledges that the study accomplishes its main objectives, “demonstrate feasibility and benefits” of training with multiple databases of MRIs with different sets of modalities. This aligns with the explicit statement by both other reviewers that the findings are interesting and important to be shared with the community.
>
> The points raised by the reviewer are recommendations for improving the clarity of some points in the paper, which we addressed with text alterations. We are glad that regardless the complexity due to the nature of the work (multiple databases, different sets of MRI modalities, multiple experimental settings for demonstrating rich insights) and the required conciseness in MIDL papers, reviewers have expressed that the work is rather clear (Reviewer NRPg “The writing is clear as is”, Reviewer#99xL “clear investigation”) but we totally agree there is room for improvement. Therefore below we describe changes we made to the paper to improve the related points:
>
>
> **R3.1: Improving explanations of Tab 2 and 4.**\
> “The explanation of the experimental comparisons in Table 2 and Table 4 in the paper is somewhat ambiguous and difficult to understand.”
>
> We agree there is some room for improvement. We have made **extensions to captions of Tables 2 and 4,  as well as in text of Sec 3**, and the changes can be seen in the updated paper **marked with red**. Thank you for the comment, we believe indeed these changes further improve clarity of the paper.
>
>
> **R3.2: Improving explanation of modality drop during training.**\
> “The strategy of Random Dropping of Modalities is not specifically elaborated”
>
> Thank you for the recommendation. We agree a few more clarifications would be useful about this point. We addressed this by making **text modifications in Section 2.1 (marked in red in updated paper)** . Text alterations are also accompanied by **improving Fig 1** that provides a complementary visual explanation of the modality drop. Thank you for the recommendation, we believe these changes help clarify this point.
>
>
> **R3.3: Making Figure 2 look nicer and clearer**\
> “The framework diagrams of the two network models in Figure 2 could be improved for better clarity and completeness.”\
> “I think that adding some colors to Figure 2 will make it more beautiful and clear.“
>
> Thank you for the recommendation. We prefer not using many colours in our papers so that they are friendly to readers that cannot see colour, and easier to print. Instead, we focused on improving the clarity of the figure by **adjusting fonts and size of text within figure 2**, as well as a clearer **separation** of the two subfigures. We believe the new version looks better, which can be inspected in the updated paper version.
>
>
> **R3.4: Improving details such as font size in figures**\
> “Details still need more attention, such as the font size of the chart should be adjusted to an appropriate size.”
>
> Thank you for your attention to details. We have revisited the paper and **made variety of such small adjustments**, such as font size in Fig 1 and Fig 2, altering some text for better readability, etc, that can be seen in the updated revised paper.
>
>
> **R3.5: Further improving clarity of Sec 2**\
> “In the methodology part of the paper, I suggest that the authors introduce the overall framework first, and then explain each module.”
>
> Thank you for the suggestion. We already try to do this with Sec 2.1. Although we cannot add a new subsection for an “overall” description of the framework due to limited space, we have now **rephrased parts of Sec 2.1** to make a smoother introduction of the framework. By improving the definitions and explanations therein, we believe these changes significantly help the reader.
>
> Thank you for the helpful review. We hope these address all your considerations.

---

### Official Review · Reviewer_99xL · 2024-02-28

**Confidence:** 3
**Preliminary Rating:** 4
**Final Rating:** 5

**Summary:**

The paper explores training a segmentation model on multiple datasets, representing different modalities and covering different pathologies. The authors investigate whether this joint training benefits the model, and if it helps the model generalise to seen pathologies on unseen modality combinations, or unseen combinations on seen pathologies. The authors find a joint UNet trained on all modalities has performance comparable to UNets trained on each modality separately. The models show some ability to segment lesions not seen during training, and provide better bases for fine-tuning on unseen datasets, when compared to training from scratch.

**Strengths:**

The work presents a clear investigation of an important topic, with a number of experiments on several datasets. I think these sorts of practical questions are important to consider.

The authors have committed to make their code available.

**Weaknesses:**

Dice isn't the only important metric when understanding segmentation performance, and I would like to see other metrics considered. One important one is the numbers of false positive/false negative segmentations we see, and understanding how exposure to new datasets with different pathologies influences this.

It would have been nice to explore the model's capability in a multi-class setting where each different pathology is assigned its own class label, in addition to the pathology vs no pathology binary setting considered here.

**Detailed Comments:**

I can't see any information on how the modality dropping is some in practice - with what probability are modalities dropped?

Were the datasets co-registered?

Table 4 could be better presented. The key thing to understand is the modalities trained on vs the modalities tested on, but not all this information is present in the Table so readers have to refer back to Table 1 to understand it.

It would be helpful to include the best possible performance (based on supervised training on the target dataset) in Table 5, to better place the results in context.

One work that also seeks to train on diverse modalities is UniverSeg. Could the authors comment on any similarities?

**Justification Of Final Rating:**

The authors have done a good job responding to all my comments as clearly as possible. I appreciate the willingness to improve the clarity of the result presentation, as well as provide further metrics. I'm happy to increase my rating.

**Justification Of The Preliminary Rating:**

This work considers an important question of interest to the community. There is limited methodological novelty here but I think the work is nonetheless a good contribution. I would like to see some more detailed presentations of the results.

**Questions To Address In The Rebuttal:**

I would like to see more detailed performance metrics, perhaps in supplementary if there is no space in the main paper. I would also like to see a clearer presentation of the results and better contextualisation, as discussed above, by showing the best possible performance on unseen datasets.

---

> ### Author Response · Authors · 2024-03-17
> **Response to R2 (#99xL) - Part 1/3**
>
> Thank you for the time taken to review the paper and the constructive feedback. \
> We are glad that the reviewer acknowledges the study investigates an “important topic”, that these “practical questions are important to consider” and that this investigation is “of interest to the community”. This aligns with similar comments from other reviewers (e.g. Reviewer #NRPg “these results could be interesting to the community”). We are also glad that the reviewer found the study rather clear (“clear investigation of an important topic”), which also aligns with other reviewers’ comments (e.g. #NRPg: “The writing is clear as is…”, #cdiz: “The overall logic of this paper is clear”), given the complexity of the study (7 databases, different pathologies, different sets of modalities, variety of experiments for rich investigation), though we do understand there is always room for improvement.
>
> The main point recommended for improvement by the reviewer is to enhance the evaluation with more metrics, and a few more requests for clarifying certain points in the text. We addressed these points by improving the text and adding material to the Appendix. We describe how we addressed each point below:
>
>
> **R2.1: Additional metrics:**\
> “Dice isn't the only important metric when understanding segmentation performance, and I would like to see other metrics considered. One important one is the numbers of false positive/false negative segmentations we see, and understanding how exposure to new datasets with different pathologies influences this.”\
> “I would like to see more detailed performance metrics, perhaps in supplementary if there is no space in the main paper.”
>
> We agree that more metrics would enhance the work. Due to limited space, we could not present more metrics in the paper. **We now provide additional metrics in the Appendix of the updated article.** Due to limited amount of time available for preparing this rebuttal and also to avoid overwhelming the Appendix with way too many tables, we prioritised calculating additional metrics for important experiments in Tables 3 and 4. We preferred calculating the commonly used metrics of Sensitivity (Recall), Precision and Average Symmetric Surface Distance (ASSD). Sensitivity and Precision quantify presence of FNs and FPs. We add ASSD, which is a distance-based metric for assessing errors, complementing the overlap-based metrics such as Dice.
>
> Tables 7 and 8 in the Appendix show the three new metrics for the same experiments that were previously analysed in Table 3. On average across all databases, MultiUnet-MD (joint training on multiple databases) achieves highest Sensitivity (low FNs) and highest Precision (low FPs) in comparison to models trained on each database separately (Unet-SD). Additionally, it achieves the best (lowest) ASSD in every single database.
>
> Tables 9 and 10 present these metrics for the same experiments described in Table 4 of the main paper (generalisation to new databases with new sets of modalities). Therein results are similar. The learning framework (trained on “All databases (drop mods)” with modality drop) improves Sensitivity in all but one experiment, Precision in all experiments, and ASSD in all experiments.
>
> *We think these clearly demonstrate the benefits of the framework from multiple perspectives.*
>
> For experiments of Table 5 we note that main DSC baselines for comparison are taken from the corresponding articles (noted as *), which do not report sensitivity/precision/ASSD, only DSC, so performing such an analysis for these experiments would be incomplete/inconsistent, which we would not like to add even to our Appendix. We believe the DSC differences are sufficiently large to convey the main message (the investigated learning paradigm does better than unsupervised anomaly segmentation) so the additional metrics do not add much further insights.\
> Finally, although we could not perform it for this rebuttal due to time limitations, we should be able to complete this analysis (sens/prec/assd) for the experiments of Table 6, to include it to the camera ready, which in this case could be interesting.

---

> ### Author Response · Authors · 2024-03-17
> **Response to R2 (#99xL) - Part 2/3**
>
> **R2.2: Further explanation of modality drop in training:**\
> “I can't see any information on how the modality dropping is some in practice - with what probability are modalities dropped?”
>
> Thank you for the good suggestion. **We improved the description of this point by making related alterations to the text in the paper in Sec 2.1.** See updated paper, changes marked with red. In short, for each training sample from database $D_i$ with $C_i$ number of modalities, we sample an integer number $n$ of modalities to drop from the range $[0, C_i-1]$ (to keep at least 1). Then we randomly choose $n$ out of the $C_i$ modalities of that sample and “drop them”, replacing them with blank images.
>
> **R2.3: Were the datasets co-registered?**
>
> No, the data were not co-registered. This is commonly avoided for brain lesion segmentation, which you can see in various studies. Even rigid co-registration can be a source of error, for example if it is suboptimal at test time. Especially for brains with large pathologies (large tumors or severe TBI, for instance) even simple rigid registration can still give suboptimal results unexpectedly and introduce errors. Therefore we do not perform it and process the databases as they originally were. We just ensure that the axes of the underlying images are aligned across databases (left/right, top/bottom, front/left), but we do not register the brains.
>
>
> **R2.4: Adding which modalities are in training of each model  in Table 4:**\
> “Table 4 could be better presented. The key thing to understand is the modalities trained on vs the modalities tested on, but not all this information is present in the Table so readers have to refer back to Table 1 to understand it.”
>
> We understand that ideally referring back to Table 1 where we show modality for each database would ideally be avoided. However, please appreciate that this study has the inherent challenge of dealing with 7 different databases, each with a different set of modalities, in variety of experimental settings, to provide a multi-dimensional analysis of the learning framework. We acknowledge that regardless the complexity, we should strive doing the best for the reader. We have tried to make each Table relatively standalone by providing long caption descriptions, and we are pleased that reviewers commented that overall the paper is rather clear (albeit improvements are of course always possible).
>
> To improve the readability of Table 4 and help the reader understand the relevant modalities, **we extended the caption and make explicit the modalities used in ATLAS , ISLES, BRATS and Tumor_2, which are the most relevant for interpreting these experiments (T1,T1c,T2,FLAIR).** The model trained on the 5 databases (BRATS, TBI, MSSEG, ATLAS, WMH, explicitly stated in caption)  has been trained to use all these modalities, as we **explicitly state in the caption**. It can, of course, use even more modalities learned from the 5 databases, such as SWI from TBI, but these are not relevant to these experiments and therefore we do not list them in the caption due to space limitation and to avoid overwhelming the reader with less relevant details. \
> We **show modalities of the *test* databases** at the top of the columns for ATLAS and Tumor_1.\
> (Note: We also tried various ways for showing the training modalities within the table too but it cannot be done elegantly due to limited horizontal space).
>
>
> **R2.5: Including upper bound by supervised training in Table 5 for context.**\
> “It would be helpful to include the best possible performance (based on supervised training on the target dataset) in Table 5, to better place the results in context.”\
> “I would also like to see a clearer presentation of the results and better contextualisation, as discussed above, by showing the best possible performance on unseen datasets.”
>
> Thank you for the suggestion. The upper bound by supervised training are the results that were already presented in Table 3, with the models “Unet-SD (all mods)”, that each has been trained with supervision on the specific database that it is evaluated on (different splits of course). We do understand, however, that it is more convenient and clear to also present them in Table 5, for clarity and completeness (initially not done to save space). **These results have now been added in Table 5 of the updated paper.** Naturally, the upper bound results by models trained on the “target” database are significantly above those achieved by any model that is not trained on the target database (as they have never seen this pathology before!). This is already explicitly stated in text (Sec 3.4) of the original paper, and is what motivates the followup investigation of fine-tuning in Table 6. *This change therefore does not affect any of the original claims* of the paper.

---

> ### Author Response · Authors · 2024-03-17
> **Response to R2 (#99xL) - Part 3/3**
>
> **R:2.6 Discuss if there is any relevance with Universeg**\
> “One work that also seeks to train on diverse modalities is UniverSeg. Could the authors comment on any similarities?”
>
> The approach of Universeg and the goal of our study are conceptually and fundamentally very different. Universeg is a method for few-shot segmentation. It assumes that each different database, every different modality and every different anatomy/pathology/target-for-segmentation is a *different task*. Therefore, every different modality is processed entirely separately both at training and test time - It is not designed for multi-modal processing, which draws relations between modalities to perform better segmentation, which is the main focus of our study. Moreover, they developed a method that in order to segment a new image, it requires as *auxiliary input* a number of pairs of images and *manual segmentations* for the target task (that guide it to segment the new image). Our study focuses on multi-modal processing, and requires no user input at test-time such as manual segmentations. Our study does investigate the benefits that a model pre-trained with the investigated framework could have for follow-up finetuning with a lot or limited labels. But we are not focusing on developing a few-shot learning method, which is the point of Universeg. We acknowledge a reader may also wonder about the relation between the 2 studies. **We have added a reference to Universeg in Sec 1 and clarify this in the text of the updated paper.** Thank you.
>
> **R2.7: Different pathologies as different classes (multi-class):**\
> “It would have been nice to explore the model's capability in a multi-class setting where each different pathology is assigned its own class label, in addition to the pathology vs no pathology binary setting considered here.”
>
> This is goes well beyond the scope of the original study - but it’s a clear objective for future studies that our own results now motivate.
> Our study aimed to show, for the first time, feasibility of training with diverse multi-modal databases. Without showing this first, which by itself is non trivial and a significant contribution (as all reviewers agree), we cannot train multi-class pathology models. Now that feasibility was shown possible in our work under the binary setting, future works can explore the follow-up problem of multi-class.\
> Also our study aimed to assess whether it is possible to make a model that can segment a new pathology. Treating all pathologies as 1 class allowed the first step in this exploration (a new unknown pathology will be part of this class). Multi-class treatment would not enable this (at least in a straightforward manner), as it could lead to the model learning separate representations per pathology class, and it is not straightforward to derive how to treat a new, previously unseen class (which class out of those in training? None corresponds). Now that our study showed for the first time that such training is feasible and beneficial, it motivates targeting this more complex approach in the future.
>
> Thank you very much for your review. We hope the above adress your points. Let us know if there is anything left we can clarify.

---

### Official Review · Reviewer_NRPg · 2024-03-05

**Confidence:** 2
**Preliminary Rating:** 4
**Recommendation:** Poster
**Final Rating:** 5

**Summary:**

This paper evaluates whether using multiple modalities and datasets can benefit MRI segmentation tasks. They perform experiments using seven different multi-model segmentation datasets to discover appropriate NN architecture and derive insights into the feasibility of improving performance on segmentation tasks. Some of the main insights from their experiments are -
- Embedding each modality separately and fusing only in penultimate layers is more susceptible to missing modalities.
- The model can learn to label pathologies even when that pathology may be only available in other modalities during training
- Moreover, the joint training with multiple modalities and datasets facilitates better finetuning of newer datasets and modalities.

**Strengths:**

- The paper is clear regarding its hypothesis, and the experiments align well to evaluate these hypotheses.
- The proposed approach is easy to implement and outperforms training with fewer datasets or modalities, clearly establishing the benefits of incorporating more datasets and modalities.

**Weaknesses:**

- It would have been nicer to compare with some of the baselines mentioned in the intro. For example, one could use the 3D foundational model (Kirillov et al.), one modality at a time, and aggregate it with majority voting or some such heuristic.
- Since the sample sizes are smaller, having error bars and showing that results hold across multiple runs would be helpful.

**Detailed Comments:**

- See weakness
- The writing is clear as is but could be improved by visually explaining the experiments. It is easy to get confused with multiple datasets & multiple modalities and such terms.
- Please provide error bars.

**Justification Of Final Rating:**

I appreciate the authors' response and the changes they have made, and hence, I am increasing my score to indicate support for this paper. Overall, this paper should be interesting to the MIDL community

**Justification Of The Preliminary Rating:**

The authors propose an interesting set of experiments to evaluate the use of multiple datasets with different modalities. The results show an obvious but interesting trend that using more datasets & modalities can be helpful with suitable training pipelines. And these results could be interesting to the community.

**Questions To Address In The Rebuttal:**

None

**Special Issue:**

No

---

> ### Author Response · Authors · 2024-03-17
> **Response to R1 (#NRPg) - Part 1/3**
>
> We thank the reviewer for the time taken to review this study and the constructive feedback. We are glad the reviewer expresses clearly that the findings would be “interesting to the community.”, which is explicitly also stated by Reviewer #99xL (“important question of interest to the community”), as well as with the assessment of Reviewer #cdiz.
> We are also glad the reviewer acknowledges that the experiments “align well to evaluate these hypotheses” of the paper (Reviewer #cdiz also acknowledges as strength that the study demonstrates the claimed contributions).
> We also are glad the reviewer found “the writing is clear”, given the complexity of the study (7 databases with different diseases and MRI modalities), though this can always be improved.
> We would like to emphasize that what this study demonstrated, that it is feasible to train a single model using data of Multimodal MRI for different pathologies, with different sets of modalities, and that there will be practical benefits, is not a-priori “obvious” neither technically nor theoretically (speculations and arguments could be made both in support and against whether it would work). Therefore, we do strongly agree that publishing these findings would be of interest to the community (emphasized by Reviewer 99xL as a strength), opening the way for further future investigations.
>
> The reviewer recommends improvements for 3 points, which we address below and the paper. Thank you for the suggestions.
>
> (We apologise for the long reply, even if the raised points were few. We preferred a longer but clear response over a shorter but less clear response :-) )
>
>
> **R1.1: Comparing with other methods, such as  SAM-Med3D**\
> “It would have been nicer to compare with some of the baselines mentioned in the intro. For example, one could use the 3D foundational model (Kirillov et al.), one modality at a time, and aggregate it with majority voting or some such heuristic.”
>
> Primary goal of our study is to investigate *feasibility* of training a *multi-modal* model (combines info from multiple input modalities) from multiple databases, each with different modalities, to explore whether there are benefits. This has never been done before and therefore there are no methods that can be straightforwardly be applied as real baselines for the task. To facilitate meaningful experimental comparisons, we already implemented 3 methods in the submitted version of the paper, that are inspired by work on missing-modalities. LFUnet is inspired by HEMIS, which can be seen as a baseline. MAFUnet is an extension made by us, and Multi-Unet is a simple practical extension of Unet with modality drop that was found effective. We showed the simplest approach worked well.
>
> Specifically about the SAM-Med3D (reviewer referenced Kirillov et al, which is the original SAM in natural images, but we guess they meant SAM-Med3D that is a 3D medical adaptation):
> - SAM-Med3D is not automatic. It is interactive and requires the user to click or draw bounding box around the object they wish to segment (e.g. the lesions in our case). Therefore it is really not meaningful comparing that with fully automatic methods like ours. We had not made this distinction clear in our related-work section because we thought it is known for SAM models, but we now agree that many readers may not be aware. **Therefore we have added text in the related-work in Sec.1 to clarify this.**
> - Moreover, SAM models use a very different backbone, a large transformer. This factor also does not facilitate meaningful comparison (even if we would create input clicks), because it would not be clear if any differences are due to the differences in architecture, size of model, or the property of training with the specific setting we explored. We also emphasize that one of the main aims of our work is to develop a simple, practical method for training the well established CNNs / Unet with multiple input modalities in the setting where modalities differ per database. Therefore the choice of extending the well established Unet is fundamental. Now that feasibility of such learning has been shown, future works can also explore transformers.
>
> (The above also apply for Kirillov et al (original SAM on non-medical images),  SAM-Med3D (non-peer reviewed and unpublished that we added for completeness) and other medical adaptations such as the 2D MedSAM [Ma et al, Segment Anything in Medical Images])
>
> **We have added text Sec.1 to clarify the above about SAM-Med3D.**

---

> ### Author Response · Authors · 2024-03-17
> **Response to R1 (#NRPg) - Part 2/3**
>
> **R1.2: Multiple runs per experiment and error bars**\
> “Since the sample sizes are smaller, having error bars and showing that results hold across multiple runs would be helpful.”
>
> Thank you for the useful suggestion. First, we emphasize that running multiple runs (seeds) per experiment is very time consuming (3D Unet, multi-modal data, multiple databases) and therefore we could re-run only specific experiments for the rebuttal. For example, training 1 model on 5 databases takes more than 4 days on a SOTA GPU, per run. Training on a single database takes 1-2 days, per run. Therefore, it is extremely expensive to re-run all experiments many times. That being said, we acknowledge that limiting potential noise is desirable.
>
> We prioritized re-running experiments where the results between approaches had smaller difference (e.g. Table 3). For tables where trends are quite clear (Tab 4, 5, 6) we will perform more runs until the camera-ready, but we do not expect results will change any of our central findings because the trends there are clear.
>
> For Table 3, during the time for the rebuttal, we completed 3 runs for each experiment in Row 1 (baseline Unet). Note that this row contains 5 database-specific models. Running 3 runs of each experiment just for this row means 15 training sessions (1-2 days each, 15-30 GPU days total just for this row). We updated Row 1 of Table 3 with these results (and corresponding entries in Table 6).\
> We are now working on updating Row 2 so that we update it towards the camera ready.\
> In the remaining rows 3 and 4 (our methods), in the originally submitted paper we were already reporting numbers obtained from averaging 3 runs per experiment. We had conducted this already to ensure we do not report noisy results by the new methods in the submission. We did not state this in the paper because Rows 1 and 2 of the same table were experiments from only 1 run (as these are baseline Unets that we have calibrated well throughout years of working with these databases, so we know results were representative), so we could not explain this difference between Rows 1-2 and 3-4 elegantly in the table’s caption without overwhelming the reader with details.\
> We are now working towards completing reruns of Row 2, and then we will state in the camera ready that whole table 3 reports averages of 3 reruns.
>
> Below we show the new results for the baseline unet (average of 3 runs) and compare it with the old results (1 run). We also compare it with Row 3 of Table 3, which shows the average performance over 3 re-runs of the MultiUnet (trained on all databases). Comparing them is the basis for the claim that it is feasible to train 1 model that does well on all databases used training training.
>
> |                                            	| MSSEG |	TBI  | WMH | BRATS |  ATLAS  | 	*Avg*  |
> |----------------------------------|-----------|--------|--------|----------|----------|-----------|
> | Unet-SD (all mods) (average x3) (new row1)  | 	0.659 | 0.515 | 0.721 |	0.908 |   0.487 | **0.658** |
> | Unet-SD (all mods)  (x1) (old row1)   | 	0.662 | 0.510 | 0.722 |	0.909 |   0.487 |  **0.658** |
> | MultiUnet-MD (drop) (average x3) (old row3)  | 	0.676 | 0.521 | 0.725 |	0.906 |   0.485 | **0.663** |
>
> Column “Avg” reports average across all 5 databases. As we see, the results by the Unet-SD do not differ much after averaging 3 runs from those in the paper for 1 run of the Unet-SD (x1, old row1). We draw the reviewer’s attention to the “Avg” column, where we see no difference is made on average by the re-runs, nor major difference for individual databases. The results do not change any of the findings and claims of the paper.
>
> We do not report standard deviations/error bars as they are not really meaningful with only 3 runs (impossible to run even more) and they will require space that we do not have in the paper. Just for re-assurance to the reviewer, the biggest difference between the extreme (worst or best performance) run and the average of 3 runs for any given database and any setting (Unet-SD or MultiUnet-MD) was 0.01 Dice, and smaller in all other cases.
>
> **Changes made to improve the paper: We updated Table 3, with the above average of x3 re-runs for row 1**. We will continue re-running experiments until the camera ready (they take time!), prioritizing Tab 3 row 2, completing the re-runs of Table 3. We will then follow with the other tables, hoping that in the camera ready all experiments will be more than 1 run. We do not expect differences that alter claims about the main hypothesis of the paper as they are already demonstrated with non-trivial improvements across multiple databases (as already acknowledged as a paper’s strength by the reviewer themself, as well as Reviewer #​​cdiz).
>
> (Please also note that as per suggestion of another reviewer, we added more metrics in the Appendix. They show consistent improvements by the studied framework, which should increase confidence in our findings.)

---

> ### Author Response · Authors · 2024-03-17
> **Response to R1 (#NRPg) - Part 3/3**
>
> **R1.3: Further improvements to explanations**\
> “The writing is clear as is but could be improved by visually explaining the experiments. It is easy to get confused with multiple datasets & multiple modalities and such terms.”
>
> We are glad the reviewer finds the explanations already clear as is, given the high complexity of the work (different pathologies and sets of MRI modalities per database, and per training setup). We agree with the reviewer there is always room for improvement.  It is impossible to add visual explanation of the experiments as there is no space for an additional figure. Instead, **we have made text modifications in various parts of Sec 2 and Sec 3, both in the main text and the captions of Tables**, clarifying further each experimental setting.
>
> Thank you for your feedback, for appreciating the paper’s value and for the useful recommendations. We believe these changes have helped clarify related points in the paper and improved consistency of the results, which we commit to continue working on till the camera ready. We hope these address sufficiently your points.

---

### Meta-Review · Area_Chair_GhJC · 2024-04-02

**Recommendation:** Accept (Poster)
**Confidence:** 5

**Metareview:**

This study explores the utility of employing multiple modalities and datasets for the segmentation of various brain diseases. Different architectural strategies were investigated, and numerous experiments were conducted to draw clear conclusions.

The strengths of this work are:
1) a huge amount experiments to assess the different generalization strategies for the segmentation of brain diseases
2) the use of 7 heterogeneous datasets with multi-modalities and various brain diseases
3) a comparison of 3 different architectures based on the baseline but yet effective U-Net

The main weakness of this work is :
1) the lack of details on the experiments, such as the dimensions on the input data for each method, the use of a patch-based strategy or not, the total number of parameters, the processing time, etc.

This study provides new insights into the understanding of generalization aspects in medical segmentation. The authors have done an excellent job of answering the reviewers' questions, which clearly improves the quality of their paper.

For all these reasons, I have decided to accept this article.

---

### Decision · Program_Chairs · 2024-04-06

Accept (Oral)